# WyckoffDiff – A Generative Diffusion Model for Crystal Symmetry

**Filip Ekström Kelvinius** [1]   **Oskar B. Andersson** [2]   **Abhijith S. Parackal** [2]   **Dong Qian** [1]   **Rickard Armiento** [2]
**Fredrik Lindsten** [1]

## Abstract

Crystalline materials often exhibit a high level of symmetry. However, most generative models do not account for symmetry, but rather model each atom without any constraints on its position or element. We propose a generative model, Wyckoff Diffusion (WYCKOFFDIFF), which generates symmetry-based descriptions of crystals. This is enabled by considering a crystal structure representation that encodes all symmetry, and we design a novel neural network architecture which enables using this representation inside a discrete generative model framework. In addition to respecting symmetry by construction, the discrete nature of our model enables fast generation. We additionally present a new metric, Fréchet Wrenformer Distance, which captures the symmetry aspects of the materials generated, and we benchmark WYCKOFFDIFF against recently proposed generative models for crystal generation. As a proof-of-concept study, we use WYCKOFFDIFF to find new materials below the convex hull of thermodynamical stability.

## 1. Introduction

Materials science is a field of research that is essential for technological advancement. With machine learning seeing success in a variety of fields, materials science is no exception. In the search for new materials, so called generative models are an attractive class of methods, and a number of models that can generate new materials have been developed (see, e.g., Park et al., 2024, for an overview). However, *crystalline* materials are often characterized by their specific symmetries, which are integral to their materials properties.

This is an aspect that only recently has been built into generative models (Jiao et al., 2024; Zhu et al., 2024; Levy et al., 2025). Instead, models without any built-in mechanisms that ensure symmetry in materials have and are still being developed (Xie et al., 2022; Jiao et al., 2023; Merchant et al., 2023; Zeni et al., 2025). As demonstrated by several works (Levy et al., 2025; Cheetham & Seshadri, 2024; Zeni et al., 2025), materials generated from methods without these explicit constraints often lack the symmetrical characteristics of materials found in databases. For example, Cheetham & Seshadri (2024) find that roughly 34 % of the materials generated by the GNoME model (Merchant et al., 2023) belong to four different space groups of which only one exists in the Inorganic Crystal Structure Database (Belsky et al., 2002) where it makes up only 1 %, and Zeni et al. (2025) mention that their MatterGen model tends to generate less symmetric structures than are present in the training data.

The symmetry of a material can be encoded in a *protostructure* description (Parackal et al., 2024, see also Section 2.1), where elements occupy Wyckoff positions in crystal structures categorized into space groups. This description avoids specifying the exact atomic coordinates, while maintaining the key structural information, which has been shown to be efficient for searching for novel stable materials by enabling an initial step where candidate crystal structures with high likelihood of being stable are identified based on the symmetry description alone. This step avoids wasting computational resources on exact coordinate calculations across all possible materials (Goodall et al., 2022). Additionally, the infinite space of continuous coordinates also opens the risk of generating degenerate materials or structures outside of the symmetry proximity. Since materials of high symmetry are generally the interesting materials to explore, generation of large sets of low symmetry materials is inefficient. Explicitly encoding symmetry could allow a generative model to only generate within a space of interesting materials of higher symmetry, allowing a symmetry-infused generative model to generate a broader variety of relevant crystalline materials compared to a generative model using exact coordinate representations.

Explicitly enforcing knowledge about symmetry in generative models for crystal structures is currently an underexplored research direction. Our approach is different from

---

[1]Department of Computer and Information Science (IDA), Linköping University, Sweden [2]Department of Physics, Chemistry and Biology (IFM), Linköping University, Sweden. Correspondence to: Filip Ekström Kelvinius or Oskar B. Andersson <filip.ekstrom@liu.se, oskar.andersson@liu.se>.

*Proceedings of the 42nd International Conference on Machine Learning*, Vancouver, Canada. PMLR 267, 2025. Copyright 2025 by the author(s).

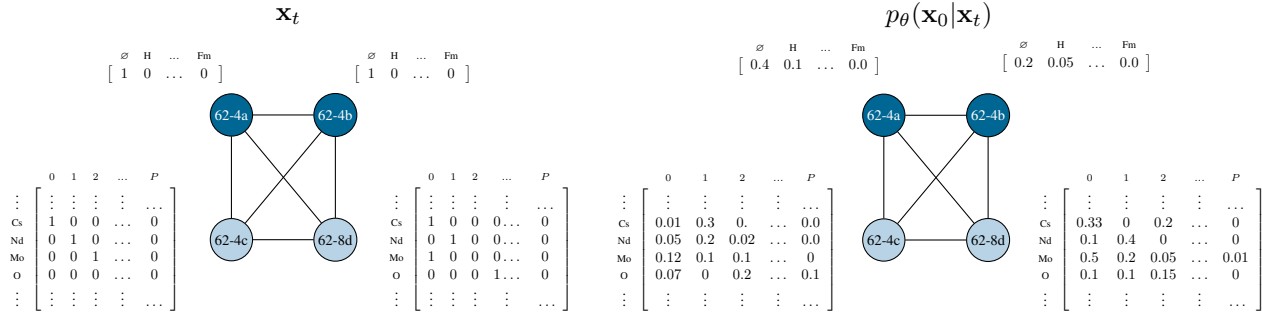

*Figure 1.* Illustration of the (graph) representation of a material used in our generative model. A material of space group 62 has four Wyckoff Positions (a, b, c, d). Two of them (a and b, dark blue) has the constraint that at most one atom can occupy the position, and we hence model that as a single variable indicating which atom type that occupies the corresponding position ($\varnothing$ denoting no atom). For the other two positions (c and d, light blue), any number of atoms can occupy the position, and we hence model this as a set of variables, one for each atom type, which indicates how many of the respective atom types that are occupying the position. To the left is the state of the material at some sampling time $t$, and to the right is the prediction of the "clean" material $\mathbf{x}_0$ made by the neural network. For all variables, there is a corresponding row in the figure, corresponding to probability vectors, and all rows hence sum to 1.

previous works in how we specifically target the generation of protostructures using a representation that enables the use of generative models *for discrete data* to generate new materials. Our method shows competitive performance against other methods on various quantitative metrics. The generated protostructures can be used as part of a machine-learning based workflow for materials discovery to find new stable crystal structures. As a proof of concept, we realize a subset of the generated protostructures into crystal structures and from this set we highlight some examples with interesting and varied chemistries ($CsSnF_6$, $NaNbO_2$, and $Ca_2PI$), which are on or below the currently known convex hull of thermodynamically stable compounds. Data and code is available online[1].

## 2. Background

### 2.1. Representing Crystals

An ideal crystalline material is commonly represented by its *crystal structure* as an infinitely repeating set of unit cells with atoms of specified *chemical elements* placed at specific atomic positions. In the unit cell, the $M$ atoms are specified by their positions $X \in \mathbb{R}^{M \times 3}$ and elements $Z \in \mathbb{Z}^M$, and the geometry of the unit cell can be specified by three lattice vectors $L \in \mathbb{R}^{3 \times 3}$. As an alternative, one can separately specify the symmetry of the atomic positions, and then specify the atomic coordinates only by precise values for the remaining *degrees of freedom*. This representation is discussed in the following.

**Protostructures**   All possible combinations of symmetries of crystal structures can be categorized into 230 *space groups* (Müller et al., 2013). The atoms, each a chemical el-

ement from the periodic table of elements, can then occupy a so called *Wyckoff position* in the crystal structure, which represents sets of points on which the symmetry operators act in a specific way. Hence, if an atom is specified to sit at a specific Wyckoff position, depending on the nature of that Wyckoff position, this declares it to reside exactly at a specific point; anywhere along a line; in a plane; or in a volume, and the symmetry operators then imply that equivalent atoms sit at a number (the *multiplicity*) of other points in the unit cell, called the *orbit*. These different Wyckoff positions are labeled using a letter from the Latin alphabet (a, b, c, etc.). The space group completely determines which Wyckoff positions that are available, as tabulated by The Volume of International Tables for Crystallography (IUCr, 2002).

In this work, we use the term *prototype* as defined for AFLOW prototype labels (Mehl et al., 2017), i.e., the combination of the spacegroup and how the Wyckoff positions are occupied by unspecified but distinct elements, without additional information about the remaining degrees of freedom for those occupied positions. In more detail, the AFLOW prototype label `ABC6_hR24_166_a_b_h` specifies first the anonymous composition `AB6C` (i.e., $AB_6C$), then the Pearson symbol `hR24`, followed by the spacegroup number `166`, and a list of Wyckoff labels for the positions occupied by the distinct elements in the anonymous formula, `a_h_b` (i.e., positions a, h, and b). Furthermore, following Parackal et al. (2024) we use the term *protostructure* to refer to a prototype where specific chemical elements are assigned to the Wyckoff positions (but where the degrees of freedom of the structure remains unspecified). Protostructures can be labeled by extended AFLOW prototype labels, e.g., `AB6C_hR24_166_a_h_b:Cs-F-Sn`, to indicate that the previously anonymous elements $A$, $B$, and $C$

---

[1] https://github.com/httk/wyckoffdiff

are `Cs-F-Sn` (Cs, F, Sn), which occupy the spacegroup 166 Wyckoff positions `a_h_b` (a, h, b)[2].

## 2.2. Diffusion Models

Diffusion models (Sohl-Dickstein et al., 2015; Ho et al., 2020; Song et al., 2021) are a type of generative models that have received tremendous interest lately. In essence, they are based on the idea of starting from a pure noise sample $\mathbf{x}_T$, which is iteratively "denoised" to end up with a "clean" sample $\mathbf{x}_0$. This denoising is enabled by viewing the data-to-noise (forward) process as a fixed Markov chain

$$q(\mathbf{x}_{0:T}) = q(\mathbf{x}_0) \prod_{t=0}^{T-1} q(\mathbf{x}_{t+1}|\mathbf{x}_t), \quad (1)$$

where $q(\mathbf{x}_0)$ is the data distribution and the transitions $q(\mathbf{x}_{t+1}|\mathbf{x}_t)$ are designed such that, for large $T$, $q(\mathbf{x}_T)$ converges to a distribution $p(\mathbf{x}_T)$ from which we can easily sample, like a Gaussian distribution in case of continuous variables. The reverse process is then parametrized as

$$p_\theta(\mathbf{x}_{0:T}) = p(\mathbf{x}_T) \prod_{t=0}^{T-1} p_\theta(\mathbf{x}_t|\mathbf{x}_{t+1}), \quad (2)$$

where $p_\theta(\mathbf{x}_t|\mathbf{x}_{t+1})$ are fitted such that $p_\theta(\mathbf{x}_t|\mathbf{x}_{t+1}) \approx q(\mathbf{x}_t|\mathbf{x}_{t+1})$. Sampling according to the reverse process will then give (approximate) samples from the data distribution $q(\mathbf{x}_0)$.

While most diffusion models have been developed for continuous data, there are also several methods designed for the discrete case (e.g., Hoogeboom et al., 2021; Austin et al., 2021; Campbell et al., 2022; Sun et al., 2023; Lou et al., 2024). Conceptually, the idea is the same, but the transitions (both in the forward and backward directions) operate on discrete state-spaces and the limiting distribution $p(\mathbf{x}_T)$ is typically chosen to factorize over the components of $\mathbf{x}_T$ to enable easy sampling. In this work we make explicit use of the method D3PM by Austin et al. (2021), which we explain in more detail in the context of our model in Section 3.

## 2.3. Related Work

**CDVAE** The Crystal Diffusion Variational Autoencoder (CDVAE) (Xie et al., 2022) is a generative model for crystal structures that combines a variational autoencoder (VAE) with a diffusion model. Generation from CDVAE starts with sampling from the VAE: a vector $z \sim \mathcal{N}(0, I)$ is sampled from which the lattice vectors $L$, the number of atoms $M$, and the initial composition are decoded. The positions of the $M$ atoms are randomly initialized, and the elements are

randomly assigned according to the decoded composition. The diffusion process then consists of denoising the positions and elements, conditioned on $z$, while keeping $L$ fixed during the full process. The positions and atoms are updated without any explicit or built-in constraints with respect to symmetries.

**DiffCSP and DiffCSP++** DiffCSP (Jiao et al., 2023) builds upon CDVAE by replacing the VAE with a diffusion model that jointly learns the lattice and coordinates, enabling more precise modeling of crystal geometry. DiffCSP++ (Jiao et al., 2024) further incorporates space group symmetry by leveraging pre-defined structural templates from the training data to learn atomic types and coordinates aligned with these templates. However, this might limit the diversity and novelty of the generated materials.

**SymmCD** To address this limitation, SymmCD (Levy et al., 2025) introduces a physically-motivated representation of symmetries as binary matrices, enabling efficient information-sharing and generalization across both crystal and site symmetries. SymmCD is related to our work in the sense that it also generates Wyckoff positions, but the approach is conceptually different: it start by sampling a number M of "representative" orbits, and then the element and the aforementioned binary representations of these are generated. We, on the other hand, will "start" from all Wyckoff positions and then generate which and how many of each element (if any) occupy each position.

**WyCryst** A similar work to ours is WyCryst (Zhu et al., 2024), which also generates only a Wyckoff-based description (and assign exact coordinates in a later step). However, this is based on a different representation than ours, and their study focuses on generation of strictly *ternary* materials while we put no such restrictions on the materials.

## 3. Wyckoff Diffusion

### 3.1. Representing a Protostructure

Given a space group $s \in G = \{1, \ldots, 230\}$, we denote the set of all possible Wyckoff positions as $L(s)$[3]. To represent a protostructure, we partition the set of Wyckoff positions into the positions without degrees of freedom (i.e., an atom occupying the position is limited to a fixed point in space) and the positions with degrees of freedom (i.e., an atom occupying the position can be positioned anywhere on a line, in a plane, or in a volume). We call these *constrained* and *unconstrained* positions, and use the notation $L_0(s) \subset L(s)$ and $L_\infty(s) \subset L(s)$ for the respective sets. Although unconstrained Wyckoff positions can virtually be occupied by any number of atoms, in our modeling, a maximum of $P$ atoms

---

[2]Note that the canonicalization of the protostructure compared to the prototype is different, due to protostructures being canonicalized based on alphabetical element order.

[3]All possible Wyckoff positions can be found in IUCr (2002).

of each type can occupy an unconstrained Wyckoff position (which means the unit cell has $P$ times the multiplicity of that Wyckoff position of such atoms). We denote $N_a$ as the largest atomic number under consideration. Both $N_a$ and $P$ can be determined from training data. Conditionally on the space group $s$, the unconstrained positions can then be represented by $\mathbf{z}^\infty \in \mathbf{M}_\infty = \{0, 1, \ldots, P\}^{|L_\infty(s)| \times N_a}$, i.e., each element $\mathbf{z}^\infty_{(i,j)} \in \{0, 1, \ldots, P\}$ is the number of atoms of type $j$ occupying the unconstrained Wyckoff position $i$. A constrained position, however, can only be occupied by 0 or 1 atoms (as the positions are restricted to a fixed point in space). Therefore, we represent the elements of the atoms occupying each of these positions as $\mathbf{z}^0 \in \mathbf{M}_0 = \{0, \ldots, N_a\}^{|L_0(s)|}$, where the value 0 corresponds to no atom occupying the position. To summarize, a protostructure can be described as the tuple[4]

$$(s, \mathbf{z}^\infty, \mathbf{z}^0) \in G \times \mathbf{M}_\infty \times \mathbf{M}_0. \quad (3)$$

### 3.2. Model Overview

Given our representation of a protostructure in Equation (3), we now aim to sample from the (unknown) distribution $p_{\text{data}}(s, \mathbf{z}^0, \mathbf{z}^\infty)$. Since the space group determines the number of Wyckoff positions, we propose to first sample a space group $s$, and then sampling the remaining variables conditioned on $s$. Using the representation $(s, \mathbf{z}^0, \mathbf{z}^\infty)$ ensures that we sample a valid material where constrained positions are occupied by at most one atom. As an estimation of the distribution of $s$, we can use the empirical training data distribution $\hat{p}_{\text{data}}(s)$, and write our model of $p_{\text{data}}(s, \mathbf{z}^0, \mathbf{z}^\infty)$ as

$$p_\theta(s, \mathbf{z}^0, \mathbf{z}^\infty) = \hat{p}_{\text{data}}(s) p_\theta(\mathbf{z}^0, \mathbf{z}^\infty | s), \quad (4)$$

where $p_\theta(\mathbf{z}^0, \mathbf{z}^\infty | s)$ is a diffusion model. We will in the next sections describe how we design $p_\theta(\mathbf{z}^0, \mathbf{z}^\infty | s)$, and when doing so, we will for simplicity use the notation $\mathbf{x}$ as the concatenation $(\mathbf{z}^0, \mathbf{z}^\infty)$, as well as keeping the conditioning on $s$ implicit. Algorithm 1 outlines the full generation of a material using WYCKOFFDIFF.

### 3.3. Discrete Diffusion

As both $\mathbf{z}^0$ and $\mathbf{z}^\infty$ are discrete variables, we will use the Discrete Denoising Diffusion Model (D3PM) (Austin et al., 2021) as our underlying diffusion model. In this framework, a datapoint is denoted as $\mathbf{x} = (x^1, \ldots, x^D)$ where each variable $x^k$ is a discrete variable, and "noise" is added independently to each variable according to a discrete Markov chain. By denoting $\mathbf{x}_t^k$ as a one-hot encoding of the $k$:th

---

[4]For ease of notation, we have omitted the dependence of $\mathbf{M}_\infty$ and $\mathbf{M}_0$ on $s$.

variable $x^k$ at sampling time $t$, the Markov forward process (cf. the general description in Section 2.2) can be written as

$$q(\mathbf{x}_{t+1}^k | \mathbf{x}_t) = \text{Categorical}(\mathbf{x}_{t+1}^k | \mathbf{p} = \mathbf{x}_t^k Q_{t+1}), \quad (5)$$

with $Q_{t+1}$ being a transition matrix, and $q(\mathbf{x}_{t+1} | \mathbf{x}_t) = \prod_{k=1}^D q(\mathbf{x}_{t+1}^k | \mathbf{x}_t)$. The matrices $Q_{t+1}$ are chosen so that the stationary distribution ($q(\mathbf{x}_T^k)$ for large $T$) is a simple distribution (we discuss this choice in Section 3.5). The variables $\mathbf{x}_t^k$ are assumed conditionally independent given $\mathbf{x}_{t+1}$ in the backward process, i.e., $p_\theta(\mathbf{x}_t | \mathbf{x}_{t+1}) = \prod_{i=1}^D p_\theta(\mathbf{x}_t^k | \mathbf{x}_{t+1})$, and as the backward distribution $q(\mathbf{x}_t^k | \mathbf{x}_{t+1}, \mathbf{x}_0^k)$ can be computed exactly, the backward process $p_\theta(\mathbf{x}_t^k | \mathbf{x}_{t+1})$ is parametrized as a marginalization over all possible $\mathbf{x}_0^k$,

$$p_\theta(\mathbf{x}_t^k | \mathbf{x}_{t+1}) = \sum_{\mathbf{x}_0^k} q(\mathbf{x}_t^k | \mathbf{x}_{t+1}, \mathbf{x}_0^k) p_\theta(\mathbf{x}_0^k | \mathbf{x}_{t+1}). \quad (6)$$

In other words, to use this framework, it is necessary to determine a suitable noise process (i.e., choosing the matrices $Q_{t+1}$), and construct and train a model which can predict the "clean" variable $\mathbf{x}_0^k$, given a noisy sample $\mathbf{x}_{t+1}$ (i.e., the model $p_\theta(\mathbf{x}_0^k | \mathbf{x}_{t+1})$).

### 3.4. WyckoffGNN – Neural Network Backbone

For the parametrization of $p_\theta(\mathbf{x}_0^k | \mathbf{x}_{t+1})$, we design a novel neural architecture, WyckoffGNN, that takes a "noisy" data point $\mathbf{x}_{t+1}$ as input, and outputs $D$ different probability vectors, where $D$ is the number of variables. This means that for the Wyckoff representation in Equation (3), the neural network needs to predict the probabilities for $D = |L_\infty(s)| \times N_a + |L_0(s)|$ different categorical distributions. To do this, we view each Wyckoff position in $L(s)$ as a node in a fully connected graph. As different space groups have different number of Wyckoff positions, using the graph representation and processing this with a graph neural network (GNN) gives us the flexibility to utilize a single model for all space groups. The GNN is used to encode each position as a vector in $\mathbb{R}^d$, and we then use a neural network to decode the vectors into the corresponding probability distributions. An illustration of this can be found in Figure 1.

**Encoding Wyckoff Positions** The encoding of Wyckoff positions starts with an initial set of vectors $\{\mathbf{h}_i^0\}_{i=1}^{|L(s)|}$, one for each Wyckoff position. These encode the atoms occupying the respective positions, i.e., $d$-dimensional vector embeddings of the atom types on the positions in $L_0$, and the number of each element on the positions $L_\infty$ (see more details in Appendix A). Additionally, we have a set of static vectors $\{\mathbf{h}_i^{\text{pos}}\}_{i=1}^{|L(s)|}$ which encode information about the position like the Wyckoff letter and the number of degrees of freedom, but also the space group $s$ and the sampling time $t$ (again, in the form of high-dimensional embedding vectors,

---
**Algorithm 1** WYCKOFFDIFF

---

**Note:** We use the notation $\mathbf{x}_t = (\mathbf{z}_t^0, \mathbf{z}_t^\infty)$. In the for-loop over $k$, if $k$ is an unconstrained position, $\mathbf{x}_0^k$ consists of $N_a$ variables and $\mathrm{MLP}(\mathbf{h}_k)$ outputs $N_a$ different probability vectors (sampled independently)

    Sample $s \sim \hat{p}_{\mathrm{data}}(s)$
    Sample $\mathbf{x}_T \sim p_\theta(\mathbf{x}_T|s)$ {*Prior distribution, e.g., assign all variables to zeros*}
    **for** $t$ in $T-1 \ldots 0$ **do**
        Encode material as $\{\mathbf{h}_k\}_{k=1}^{|L(s)|} = \mathrm{GNN}(s, \mathbf{x}_{t+1})$
        **for** $k$ in $1 : |L(s)|$ **do**
            $p_\theta(\mathbf{x}_0^k|\mathbf{x}_{t+1}, s) = \mathrm{Cat}\big(\mathbf{x}_0^k; \mathbf{p} = \mathrm{MLP}(\mathbf{h}_k)\big)$
            Compute $p_\theta(\mathbf{x}_t^k|\mathbf{x}_{t+1}, s)$ according to Equation (6)
            Sample $\mathbf{x}_t^k \sim p_\theta(\mathbf{x}_t^k|\mathbf{x}_{t+1}, s)$
        **end for**
    **end for**
    **return** $s, \mathbf{x}_0$

---

see Appendix A). We then design the $l$:th update of the vectors as first concatenating $\mathbf{h}_i^{l-1}$ with its corresponding $\mathbf{h}_i^{\mathrm{pos}}$, and then one layer of a message-passing neural network (Gilmer et al., 2017) where first, for each Wyckoff position, a message $\mathbf{m}_i^l$ is computed as $\mathbf{m}_i^l = \sum_{j=1}^{|L(s)|} M_l(\mathbf{w}_i, \mathbf{w}_j)$, where $\mathbf{w}_i$ and $\mathbf{w}_j$ are the aforementioned concatenation of vectors. The message $\mathbf{m}_i^l$ is hence an aggregation of messages sent between pairs of Wyckoff positions, and the purpose is to propagate information about the full material. As we do not have an inherent graph but rather assume a complete graph, we construct a message function $M_l(\cdot, \cdot)$ inspired by Bronstein et al. (2021, chapter 5.4) where we use two multilayer perceptrons (MLPs, or fully connected neural networks). One MLP takes in the neighboring vector $\mathbf{w}_j$ and outputs a new vector $\mathbf{w}_j' = \mathrm{MLP}_\phi(\mathbf{w}_j)$, while the other takes as input a concatenation of $\mathbf{w}_i$ and $\mathbf{w}_j$ and outputs a scalar $a_{i,j} = \mathrm{MLP}_\theta(\mathrm{cat}(\mathbf{w}_i, \mathbf{w}_j))$, which is multiplied with $\mathbf{w}_j'$, i.e.,

$$M_l(\mathbf{w}_i, \mathbf{w}_j) = a_{i,j}(\mathbf{w}_i, \mathbf{w}_j)\mathbf{w}_j'(\mathbf{w}_j). \tag{7}$$

The message $\mathbf{m}_i^l$ is hence a linear combination of transformations of the neighbor vectors $\mathbf{z}_j$. This message is then added to the current vector, so that the updated vector representation becomes $\mathbf{h}_i^l = \mathbf{h}_i^{l-1} + \mathbf{m}_i^l$. Performing such updates $N$ times (i.e., a neural network with $N$ layers), we obtain our encoded positions as the vector representations $\{\mathbf{h}_j^N\}_{j=1}^{|L(s)|}$. Algorithms describing the GNN layer and the message function together with more details on hyperparameter choices can be found in Appendix A.

**Decoding the Probabilities**  When we have obtained the encodings $\{\mathbf{h}_i^N\}_{i=1}^{|L(s)|}$ of the Wyckoff positions, we need to decode these into vectors of probabilities. For constrained Wyckoff positions, $L_0$, this corresponds to probabilities over which atom type (if any) that is occupying the position. For the unconstrained Wyckoff positions $L_\infty$, it instead corresponds to, for each atom type, the probabilities over the number of atoms of the corresponding atom type that occupies this position. As the output differs between these two types of positions, we use two different MLPs for the decoding. For the constrained positions, an MLP takes as input the representation $\mathbf{h}_i^N$ and outputs a single vector of probabilities over atomic numbers, where we use 0 as "no atom" and only consider the atomic numbers 1 to $N_a = 100$, as there are no training data points involving higher atomic numbers. For the unconstrained positions, an MLP instead outputs $N_a$ different probability vectors over number of atoms, one for each atom type. Again, we use a truncated range of 0 to $P = 54$ based on training data. An algorithm outlining the full forward-pass of the neural network can be found in Algorithm 2 in the appendix, together with more details in Appendix A.

**Training**  To train our neural network, we start by sampling a time $t$ from the discrete uniform distribution $\mathrm{Uniform}([1, \ldots, T])$. Then, to sample $\mathbf{x}_t \sim q(\mathbf{x}_t|\mathbf{x}_0)$, we sample $\mathbf{x}_t^k \sim q(\mathbf{x}_t^k|\mathbf{x}_0) = \mathrm{Categorical}(\mathbf{p} = \mathbf{x}_0^k \overline{Q}_t)$ independently for each $k \in \{1, \ldots, D\}$, where $\overline{Q}_t = Q_1 \cdots Q_{t-1} Q_t$, and the choice of $Q_t$ is described in Section 3.5. The neural network takes as input this noisy sample $\mathbf{x}_t$, and as in DiGress (Vignac et al., 2022), we optimize the cross-entropy between the true sample $\mathbf{x}_0$ and the predicted distribution $p_\theta(\mathbf{x}_0|\mathbf{x}_t)$. We also tried the variational objective by Austin et al. (2021), but the large state spaces made it unfeasible to fit into GPU-memory.

### 3.5. The Choice of $Q_t$

Austin et al. (2021) proposes a few different choices of $Q_t$. In our work, we use a matrix of the form

$$Q_t = (1 - \beta_t)I + \beta_t \mathbb{1}\mathbf{m}^T, \tag{8}$$

where $\beta_t$ is given by some user-defined schedule, $\mathbb{1}$ is a vector of ones, and $\mathbf{m}$ is a vector of probabilities. With this transition matrix, a variable stays in its current state with probability $1 - \beta_t$, and with probability $\beta_t$ it transitions to a new state sampled from a $\mathrm{Categorical}(\mathbf{p} = \mathbf{m})$ distribution. This is a general form for which the choice $\mathbf{m} = \mathbb{1}/D$ gives rise to D3PM-uniform by Austin et al. (2021). In this general form, for large $T$, the limiting distribution $q(\mathbf{x}_T^k)$ becomes $\mathrm{Categorical}(\mathbf{p} = \mathbf{m})$, and sampling from D3PM hence starts by sampling each variable $\mathbf{x}_T^k$ from this distribution. Although using the uniform distribution could work, in case the data is very "sparse", for example in our case where most of the elements in the matrix representation in Section 2.1 are 0, using the uniform distribution as the limiting distribution could require many generation steps just to find the correct level of "sparseness". Vignac et al.

*Table 1.* Results on the material generation task. All metrics are computed for 10 000 samples, and we present averages and standard deviations for three models trained with different seeds. To compute FWD, the training set was subsampled to contain an equal number of samples. In the case of novel materials, 10 000 novel materials have been generated. The different options for WYCKOFFDIFF indicates the different prior (limiting) distributions. *Models trained only for 100 instead of 1 000 epochs. **SymmCD is somewhat unstable and produces materials with `NaN` values ($\sim 4\%$ of the materials), while WYCKOFFDIFF-uniform produces a few materials with 0 atoms ($\lesssim 0.05\%$), and we therefore discard these, meaning the numbers for these models are slightly biased (however, the numbers are still computed on 10k samples).

| | MODEL | FWD ↓ | NOV. ↑ (%) | UNIQ. ↑ (%) | NOVEL | | NOV./MIN. ↑ |
| | | | | | FWD ↓ | UNIQ. ↑ (%) | |
|---|---|---|---|---|---|---|---|
| | CDVAE | $41.9 \pm 2.69$ | $99.4 \pm 0.06$ | $99.9 \pm 0.00$ | $41.8 \pm 2.60$ | $99.9 \pm 0.00$ | 71 |
| | DIFFCSP++ | $0.83 \pm 0.14$ | $48.4 \pm 0.56$ | $98.4 \pm 0.12$ | $5.15 \pm 0.17$ | $98.7 \pm 0.06$ | 46 |
| | SYMMCD** | $1.47 \pm 0.29$ | $52.3 \pm 1.21$ | $98.4 \pm 0.12$ | $4.53 \pm 0.65$ | $98.9 \pm 0.06$ | 115 |
| OURS | WYCKOFFDIFF-UNIFORM** | $2.29 \pm 0.15$ | $40.2 \pm 0.40$ | $98.2 \pm 0.15$ | $13.71 \pm 0.61$ | $98.0 \pm 0.12$ | 159 |
| | WYCKOFFDIFF-MARGINAL* | $1.65 \pm 0.07$ | $55.7 \pm 1.95$ | $98.6 \pm 0.21$ | $6.71 \pm 0.94$ | $98.9 \pm 0.06$ | - |
| | WYCKOFFDIFF-MARGINAL | $0.55 \pm 0.05$ | $31.4 \pm 1.46$ | $98.0 \pm 0.12$ | $4.57 \pm 0.45$ | $97.6 \pm 0.23$ | 125 |
| | WYCKOFFDIFF-ZEROS* | $1.03 \pm 0.24$ | $54.9 \pm 2.54$ | $98.8 \pm 0.17$ | $5.39 \pm 0.22$ | $99.3 \pm 0.15$ | - |
| | WYCKOFFDIFF-ZEROS | $0.48 \pm 0.02$ | $30.2 \pm 0.97$ | $98.1 \pm 0.23$ | $4.34 \pm 0.56$ | $98.1 \pm 0.15$ | 119 |

(2022) propose to use the empirical marginal distribution instead of the uniform distribution as $\mathbf{m}$. As we show in the experiments section, we find that using a marginal distribution, or a Dirac distribution at zero for all variables (i.e., starting from a material without any atoms at all), greatly improves the performance compared with using the uniform distribution.

### 3.6. Evaluation Metric – Fréchet Wrenformer Distance

To evaluate a generative model, we strive to find a way of projecting materials into some lower-dimensional space, and draw conclusions about the difference between generated materials and real materials in this space. To do this, we take inspiration from the Fréchet Inception distance used for image generation (Heusel et al., 2017), and propose the metric Fréchet Wrenformer distance (FWD). This metric computes the Wasserstein distance between Gaussian distributions fit with embeddings of the generated materials and training set, respectively, extracted from the pretrained Wrenformer (Riebesell et al., 2024), which adapts the GNN-based model by Goodall et al. (2022) to a Transformer architecture (Vaswani et al., 2017) and is distributed with the `aviary` software[5]. The FWD metric aims to capture the similarities of the generated materials with the training materials, while being invariant to exact geometry as the Wrenformer only takes into account the protostructure of the material. Similar developments have been done for chemical (Fréchet ChemNet distance, FCD (Preuer et al., 2018)) and biological (Fréchet Biological distance, FBD (Stark et al., 2024)) applications.

## 4. Numerical Evaluations

### 4.1. FWD, Novelty, and Uniqueness

The quantitative evaluation of our models uses the WBM dataset[6] (Wang et al., 2021) created by substitution of chemical elements in the crystal structures available from the Materials Project (MP) (Jain et al., 2013) to generate a total of 257k materials. We set aside 10k+10k materials as validation and test sets. We start by comparing WYCKOFFD­IFF with CDVAE (Xie et al., 2022), DiffCSP++ (Jiao et al., 2024), and SymmCD (Levy et al., 2025) as they constitute examples of models that to different degrees model crystal symmetry. Implementation details of these baseline methods can be found in Appendix B. It should be noted that we encountered some numerical issues during generation with SymmCD, resulting in `NaN` values, and we chose to discard these failed materials ($\sim 4\%$ of samples, see more details in Appendix B). We also found that using WYCK­OFFDIFF with uniform initialization can produce a small amount ($\lesssim 0.05\%$) of "void" materials with 0 atoms, which we also discarded.

As the focus of our work is on the generation of protostructures and the compared methods all generate full geometries, we convert these materials to AFLOW protostructures (Mehl et al., 2017) using `aviary`[5], with default tolerance parameters. For all methods, we generate 10 000 protostructures and compute the FWD, novelty (Nov., the fraction of generated protostructures not present in the training set), and uniqueness (Uniq., fraction of unique protostructures among the generated). We discuss validity in Appendix D. The re-

---

[5] https://github.com/CompRhys/aviary/tree/main

[6] We provide an experiment on Carbon24 (Pickard, 2020) in Appendix E

*Table 2.* The number of unique and novel prototypes among 10 000 novel protostructures.

| MODEL | # UNIQUE & NOVEL PROTOTYPES |
|---|---|
| CDVAE | $2083 \pm 61$ |
| DIFFCSP++ | $527 \pm 39$ |
| SYMMCD | $780 \pm 49$ |
| WYCKOFFDIFF-UNIF. | $1214 \pm 32$ |
| WYCKOFFDIFF-MARG. | $733 \pm 11$ |
| WYCKOFFDIFF-ZEROS | $1175 \pm 80$ |

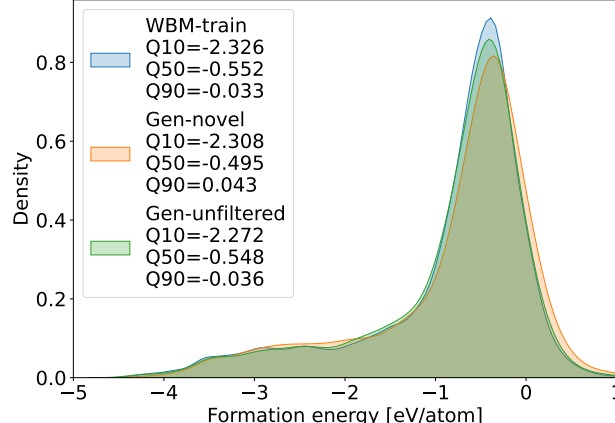

*Figure 2.* Distribution of formation energies predicted by Wren for WYCKOFFDIFF-zeros generated (unfiltered) protostructures and novel protostructures, relative to the training set. Q10, Q50,and Q90 are the 10th, 50th, and 90th percentiles respectively.

sults are presented in Table 1. It should be noted that, in this discrete setting, we do not expect the novelty to be 1 even for a "perfect model". However, in a practical materials discovery setting we are mainly interested in the novel materials and, since FWD is a metric that benefits from sampling materials from the training set, we also compute FWD and uniqueness among only novel materials. To do this, we generate enough materials so that we have obtained 10 000 novel protostructures from all methods. To simulate the computational cost if applying the postprocessing step of filtering out novel materials, we provide the number of novel materials per minute (nov./min). u

From Table 1, we first conclude that CDVAE, which does not incorporate any knowledge about symmetry, generates materials that are very dissimilar to the training distribution, as indicated by the very high FWD. As FWD measures similarity based on protostructures, the high value is likely to be due to the inability to capture the symmetry properties of materials. By examining the distribution of space groups, we find that 36% of the materials generated by CDVAE are from space group 1, and >90% are in space gropus 1-15, while the corresponding numbers for WBM are 0.3% (SG 1) and 13% (SG 1-15). Similar results were found by (Levy et al., 2025).

We also notice that the choice of initial distribution in WYCKOFFDIFF makes a big difference, and using the uniform distribution severely underperforms compared to initializing from the marginal distribution, or with completely empty materials. This highlights that even if the model is supposed to "denoise", starting from something that is closer to the actual data plays a big role. Compared to the baselines, we notice that the novelty for WYCKOFFDIFF is somewhat lower, which seems to be connected with training time: numbers for models trained with only 10 % of the number of steps shows a higher novelty, indicating that the model is "memorizing" the training distribution. However, looking at sampling speed, WYCKOFFDIFF is much faster as it does not generate full geometries, and hence, even if the novelty is lower, we produce more novel materials with the same amount of computation time, and we could view

this "novelty filer" as part of the generative procedure. Additionally, when computing FWD on only novel materials, WYCKOFFDIFF outperforms all baselines, indicating that even if the protostructures are novel, they are to a larger extent faithful to the training distribution.

### 4.2. Prototype Uniqueness

In Section 4.1, materials were classified as different if their protostructures were different. Now, we consider only the prototypes to evaluate the models' abilities to generate structural novelty. Among the 10 000 novel protostructures, we count the number of unique and novel prototypes and present this in Table 2. We see that our model indeed generates new prototypes, which highlights that it is not merely learning a "substitution-algorithm", where it learns to use an already know structural template (i.e., the prototype) and just replace the elements. We also see that only CDVAE performs better in this regard, but as CDVAE has no restrictions in its generation, this is expected. However, when comparing to DiffCSP++ and SymmCD which do take symmetry into account, WYCKOFFDIFF produces significantly higher number of unique and novel prototypes, showing its promise as a general generative model for crystal structures.

### 4.3. Wren Energies

To further investigate the protostructures generated by WYCKOFFDIFF and get a sense of their usefulness, we compare the formation energies (i.e., the energy required to form a material from the pure elements, see Appendix F for more details) of the generated protostructures with those of the training set. To compute the formation energies, we rely on the same pretrained Wrenformer model as used for FWD

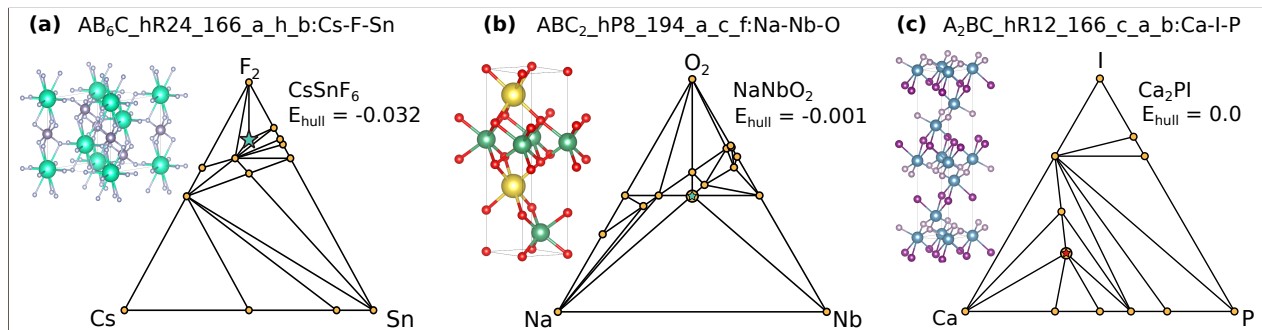

*Figure 3.* Selection of a three examples out of WYCKOFFDIFF generated crystal structures close to or below the convex hull of WBM and Materials Project (MP). Displaying the energy above hull $E_{hull}$ [eV] relative to the convex hull of WBM and MP combined. (a) has a formation energy of $E_{form} = -2.610$ the resulting in $E_{hull}$ being negative distinctly below hull. In comparison with the convex hull structure (a) is indeed below the hull, highlighted with the green star in the phase diagram. (b) has a formation energy of $E_{form} = -2.537$, resulting in a negative $E_{hull}$ but insignificantly far from the hull. (c) has a formation energy of $E_{form} = -1.422$ which makes the $E_{hull}$ approximately zero. Comparing (b) and (c) with the convex hull shows that the structures are on the hull, indicated by the smaller stars.

(see Section 3.6), which can predict the formation energy only given a protostructure. Figure 2 shows histograms of formation energies of protostructures generated by the zeros-initialization model. We see that the materials in general follow the same distribution as the training set, where the novel materials have a slight shift towards higher energies. A possible explanation is that the training data, ultimately derived from structures seen in experiments, samples the lowest energy structures thoroughly enough that the filtering on novel materials rejects more lower energy structures than higher energy ones. This further suggests the ability of WYCKOFFDIFF to generate protostructures that are also physically plausible. We see overall the same results for the distributions for the other versions of WYCKOFFDIFF, and present those in Appendix C.

## 5. Materials Discovery Using WYCKOFFDIFF

We now demonstrate how WYCKOFFDIFF fits into a materials discovery pipeline. Starting with a generation of 20 000 novel crystal structures, 10 000 from each of two WYCKOFFDIFF models (WYCKOFFDIFF-zeros and a previous iteration of WYCKOFFDIFF-marginal; see Appendix G.3), we extract structures with chemical elements that are not noble gasses and where the underlying computational methods used for the training data are known to be more reliable, i.e., elements from the s-, p-, and d-blocks of the periodic table of elements.

We then realize the resulting 12 650 protostructures into crystal structures by a process where we first semi-randomly assign values to the degrees of freedom of the Wyckoff positions using the Pyxtal library (Fredericks et al., 2021) using the implementation in aviary[5]. Subsequently, we

use the interatomic potential MACE[7] (Batatia et al., 2023) to perform a constrained relaxation where the energy is minimized while the symmetries set by the protostructure are retained. We repeat this process of realizing and relaxing crystal structures until the two lowest energies seen lies within a small cutoff of $0.01$ eV/atom. The lowest energy found is taken as our computationally predicted energy of the material generated by WYCKOFFDIFF. As is common in materials science, this energy is converted into a formation energy by for each atom subtracting the corresponding energy per atom from a representative elemental solid.

Low formation energies are only indirectly related to stability; the thermodynamically stable material at a composition is the one with the lowest formation energy compared to all alternative competing phases and linear combinations of phases, which spans the so called convex hull of thermodynamic stability (see, e.g., Bartel et al. (2020) and Appendix F for more details). However, given the indirect relationship, we selected 200 structures with the lowest formation energies to investigate further. We used the high-throughput toolkit (httk) (Armiento, 2020) to recalculate them with density functional theory (DFT) using the VASP electronic-structure software (Kresse & Hafner, 1994) and evaluated their stability relative to the known convex hull from all materials in the MP (Jain et al., 2013) and WBM (Wang et al., 2021) databases (further details in G.2).

Out of the 200 selected materials, we highlight three hand-picked examples with interesting chemistries ($CsSnF_6$, $NaNbO_2$, and $Ca_2PI$), shown in Figure 3 in their respective composition phase diagrams generated using pymatgen (Ong et al., 2013). The DFT results for these generated materials confirm them to be stable; one is distinctly below, and

---

[7]https://github.com/ACEsuit/mace-mp/releases/tag/mace_mpa_0

the other two are *on*, the convex hull. Hence, the generated structure for $CsSnF_6$ is clearly a new predicted material not present in MP or WBM. The other two materials, $NaNbO_2$, and $Ca_2PI$, already exist in MP (i.e., they are part of the known convex hull and therefore on it), and can be traced to experimental works (Roth et al., 1993; Hadenfeldt & Herdejürgen, 1988). These are thus explicit examples of WYCKOFFDIFF recreating materials outside of its training set (WBM), which are experimentally confirmed to exist. These results substantiate the ability of the model to generate materials that are physically reasonable. Furthermore, our investigation of the 200 selected materials finds seven other fluorides confirmed by DFT to be distinctly below the known convex hull from WBM and MP (details presented in Appendix G.1, Table 6). The over-representation of new stable fluorides in this set of 200 materials is likely due to that our proof-of-concept methodology of extracting the smallest, i.e., most negative, formation energies may bias towards this chemistry, rather than being a feature of the model.

## 6. Discussion & Conclusions

In this paper we propose WYCKOFFDIFF, a novel generative model which leverages a new representation of the symmetrical aspects of materials together with a novel neural network architecture and discrete diffusion to generate new protostructures. Although obtaining the full material requires extra steps, viewing the protostructure and the full geometry as separate processes opens up the possibility of using models tailored for each respective task, and use of computational effort where it is most needed. As we highlight with our proof-of-concept materials discovery pipeline in Section 5, the precise geometry can be uncovered via a pretrained generally applicable interatomic potential such as MACE, only for the most promising materials. WYCKOFFDIFF shows competitive performance compared to the current state-of-the-art both in terms of novel generated materials/min, structural novelty, and agreement with the data distribution based on the newly proposed Fréchet Wrenformer Distance.

## Acknowledgments

This work was partially supported by the Knut and Alice Wallenberg Foundation (KAW) via the Wallenberg AI, Autonomous Systems and Software Program (WASP) and the Wallenberg Initiative Material Science for Sustainability (WISE) through the joint WASP-WISE project *Generative AI models for property to structure materials prediction*.

F.E.K., D.Q., and F.L. further acknowledge support from the Swedish Research Council (VR) grant no. 2020-04122, 2024-05011, KAW project 2020.0033, and the Excellence Center at Linköping–Lund in Information Technology (EL-LIIT). R.A, O.B.A, and A.S.P acknowledge support from the Swedish Research Council (VR) grant no. 2020-05402 and the Swedish e-Science Centre (SeRC).

Parts of the computations were enabled by the Berzelius resource provided by the Knut and Alice Wallenberg Foundation at the National Supercomputer Centre (NSC). Other computations performed at NSC and Chalmers Centre for Computational Science and Engineering (C3SE) were enabled by resources provided by the National Academic Infrastructure for Supercomputing in Sweden (NAISS), partially funded by the Swedish Research Council through grant agreement no. 2022-06725.

## CRediT Authorship Contribution Statement

**Filip Ekström Kelvinius:** Data curation, Formal analysis, Investigation, Methodology, Project administration, Software, Validation, Visualization, Writing – original draft, Writing – review & editing **Oskar B. Andersson:** Data curation, Formal analysis, Investigation, Methodology, Software, Validation, Visualization, Writing – original draft, Writing – review & editing **Abhijith S. Parackal:** Data curation, Formal analysis, Investigation, Methodology, Software, Validation, Visualization, Writing – original draft, Writing – review & editing **Dong Qian:** Formal analysis, Investigation, Methodology, Software, Validation, Visualization, Writing – original draft, Writing – review & editing **Rickard Armiento:** Conceptualization, Funding acquisition, Supervision, Validation, Writing – original draft, Writing – review & editing **Fredrik Lindsten:** Conceptualization, Funding acquisition, Supervision, Writing – original draft, Writing – review & editing

## Impact Statement

This paper presents work whose goal is to advance the field of Machine Learning. There are many potential societal consequences of our work, none which we feel must be specifically highlighted here.

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

# A. WyckoffGNN Details

## A.1. Architecture

Here we give some more details on our neural network backbone, WyckoffGNN. As mentioned in the main text, it is based on the message-passing neural network framework (Gilmer et al., 2017), where each node in a graph is represented by a vector $\mathbf{h}_i^l$, and each layer corresponds to an update of this representation according to

$$\mathbf{m}_i^{l+1} = \sum_{j \in \mathcal{N}(i)} M_l(\mathbf{h}_i^l, \mathbf{h}_j^l), \tag{9a}$$

$$\mathbf{h}_i^{l+1} = U_l(\mathbf{h}_i^l, \mathbf{m}_i^{l+1}). \tag{9b}$$

Algorithm 2 describes the full pass through the network. It makes use of Embedding() layers which maps discrete features, like the atom types or number of atoms of a certain atom type, to vectors in some vector space $\mathbb{R}^d$, and Linear() which are affine maps of vectors in $\mathbb{R}^{d_{\text{in}}}$ to $\mathbb{R}^{d_{\text{out}}}$, i.e., $\text{Linear}(\mathbf{x}) = \boldsymbol{W}\mathbf{x} + \mathbf{b}$. The embedding of the number of atoms embeds the number of atoms of each atom type in $\mathbf{z}^\infty$ into a scalar which are concatenated and then processed by a linear layer such that all initial representations $\mathbf{h}^0$ of all Wyckoff positions are of the same dimension.

Algorithm 3 describes the update of the hidden representations as in Equation (9). As we are working on a fully connected graph, the sum over the neighbors is over all positions. In our case, the input to $M_l$ is not the hidden representations $\mathbf{h}_i^l$ and $\mathbf{h}_j^l$, but concatenations of the hidden representations and its corresponding *position vector* $\mathbf{h}_i^{\text{pos}}$ which contains some general information of the Wyckoff position like the number of degrees of freedom, the letter, but also the space group and sampling timestep $t$. Algorithm 4 outlines how $M_l$ is computed.

## A.2. Choice of $\beta_t$

As a scheduler for $\beta_t$, we used the cosine scheduler by Hoogeboom et al. (2021). By defining $\alpha_t = 1 - \beta_t$ and $\bar{\alpha}_t = \prod_{s=1}^t \alpha_s$, we choose $\beta_t$ such that

$$\bar{\alpha}_t = \cos\left(\frac{t/T + s}{1 + s}\frac{\pi}{2}\right), \tag{10}$$

with $s = 0.008$.

---

**Algorithm 2** Full GNN forward pass

---

**Input:** Spacegroup $s$, positions with no constraints $\mathbf{z}^\infty \in \{0, 1, \ldots, P\}^{L_\infty(s) \times N_a}$, positions with no degrees of freedom $\mathbf{z}^0 \in \{0, \ldots, N_a\}^{L_0(s)}$, number of DOFs $\mathbf{x}_{\text{dof}} \in \{0, \ldots, 3\}^{L(s)}$, Wyckoff letters $\mathbf{x}_{\text{letter}} \in$, timestep $t$
    **Output:** Probability vectors $\mathbf{p}^\infty \in \Delta_P^{L_\infty \times N_a}$ and $\mathbf{p}^0 \in \Delta_{N_a}^{L_0}$, where $\Delta_n$ is the $n$-simplex
    $\mathbf{h} \leftarrow \text{stack}(\text{Embedding}(\mathbf{z}^0), \text{Linear}(\text{Embedding}(\mathbf{z}^\infty)))$
    $\mathbf{h}_{\text{pos}} \leftarrow \text{Embedding}(\mathbf{x}_{\text{dof}}) + \text{Embedding}(\mathbf{x}_{\text{letter}}) + \text{Embedding}(s) + \text{Embedding}(t)$
    **for** layer in GNN_layers **do**
      $\mathbf{h} \leftarrow \text{layer}(\mathbf{h}, \mathbf{h}_{\text{pos}})$ {*Algorithm 3*}
      $\mathbf{h} \leftarrow \text{activation}(\mathbf{h})$
    **end for**
    $\mathbf{p}^\infty \leftarrow \text{MLP}_\theta(\mathbf{h}[\mathbf{x}_{\text{dof}} \neq 0])$
    $\mathbf{p}^0 \leftarrow \text{MLP}_\phi(\mathbf{h}[\mathbf{x}_{\text{dof}} = 0])$
    **return** $\mathbf{p}^0, \mathbf{p}^\infty$

---

**Algorithm 3** GNN layer forward pass. All operations are for $i = 1, \ldots |L(s)|$, where $|L(s)|$ is the number of Wyckoff positions for the spacegroup $s$

---

**Input:** Node features $\mathbf{h}^l = (\mathbf{h}_1^l, \ldots, \mathbf{h}_{L(s)}^l)$, position specific embeddings $\mathbf{h}_{\text{pos}}$
**Output:** Updated features $\mathbf{h}^{l+1}$

$\mathbf{w} \leftarrow \text{cat}(\mathbf{h}, \mathbf{h}_{\text{pos}})$
$\mathbf{m}_i^{l+1} \leftarrow \sum_{j=1}^{|L(s)|} M_l(\mathbf{w}_i, \mathbf{w}_j)$ {$M_l$ from Algorithm 4. Complete graph, hence sum over all other positions.}
$\mathbf{h}_i^{l+1} \leftarrow \mathbf{h}_i^l + \mathbf{m}_i^{l+1}$ {$U_l$, a simple skip connection}
**return** $\mathbf{h}^{l+1}$

---

**Algorithm 4** GNN message, $M_l(\mathbf{w}_i, \mathbf{w}_j)$ in Equation (9)

---

**Input:** Node features $\mathbf{w}_i, \mathbf{w}_j$
**Output:** Message $\mathbf{m}_{i,j} = M(\mathbf{w}_i, \mathbf{w}_j)$

$\mathbf{v}_{i,j} \leftarrow \text{cat}(\mathbf{w}_i, \mathbf{w}_j)$
$a_{i,j} \leftarrow \text{MLP}_\theta(\mathbf{v}_{i,j})$ {Scalar}
$a_{i,j} \leftarrow \text{softmax}_j(a_{i,j})$ {Will depend on other features, so cannot do this before computing $a_{i,j}$ for all $j$}
$\mathbf{m}_{i,j} \leftarrow a_{i,j} \text{MLP}_\phi(\mathbf{w}_j)$
**return** $\mathbf{m}_{i,j}$

---

*Table 3.* Hyperparameters used for WyckoffDiff

| | PARAMETER | VALUE |
|---|---|---|
| GENERAL | MAX. TIMESTEP $T$ | 1 000 |
| | MAX. ATOM NUMBER $N_a$ | 100 |
| | MAX. NUM ATOMS OF AN ELEMENT $P$ | 54 |
| GNN | NUMBER OF GNN LAYERS, $N$ | 3 |
| | DIMENSION OF $\mathbf{h}_i^l$ | 256 |
| | DIMENSION OF $\mathbf{h}_i^{\text{POS}}$ | 16 |
| | ACTIVATION FUNCTION | SiLU (SEE EQUATION (11)) |
| MLPS, GENERAL | NUMBER OF HIDDEN LAYERS | 2 |
| | ACTIVATION | SiLU |
| MLPS IN $M_l$ | HIDDEN DIMENSION | $2(\text{DIM}(\mathbf{h}_i^l) + \text{DIM}(\mathbf{h}_i^{\text{POS}})) = 544$ |
| PROBABILITY DECODING MLPS | HIDDEN DIMENSION | $2\text{DIM}(\mathbf{h}_i^l) = 512$ |
| TRAINING | OPTIMIZER | ADAMW (LOSHCHILOV & HUTTER, 2019) |
| | LEARNING RATE | $2 \cdot 10^{-4}$ |
| | BATCH SIZE | 256 |
| | NUMBER OF EPOCHS | 1000 |

### A.3. Hyperparameters and Training Details

Training of a model required approximately 38 hours on a single NVIDIA A100. Hyperparameters for WYCKOFFDIFF and its training can be found in Table 3. The activation function SiLU (Ramachandran et al., 2017) is given by[8]

$$\text{SiLU}(x) = x \frac{\exp(x)}{1 + \exp(x)}. \tag{11}$$

We did not perform any hyperparameter search.

---

[8]See also, e.g., https://pytorch.org/docs/stable/generated/torch.nn.SiLU.html

### A.4. A Note on Scalability

A bottleneck in our method is that we are operating on complete graphs, meaning that for space groups with many positions, the number of edges in the graph increases quickly. On the other hand the data dimensionality is fixed for a certain space group, and more atoms in the unit cell does not change that. E.g., in Figure 1, the number of Cs atoms occupying the "c" position is represented by an integer, so increasing this from 0 to, say, 4, doesn't affect the dimensionality of the data. Increasing the size of the set of elements in the materials (e.g., increasing $N_a$) and increasing the maximum number of atoms occupying an unconstrained position (i.e., $P$) will add additional computational overhead as, e.g., the backward transition requires summing over all possible values of a variable.

## B. Implementation Details of Compared Methods

For all methods, we used the official public implementations[9][10][11] and we train all methods for 1 000 epochs. We specify further details below.

### B.1. CDVAE

For CDVAE, we used the hyperparameters used for the MP20 dataset by the original authors, except for the learning rate which we lowered to $2 \cdot 10^{-4}$, as the default value led to instabilities in the training.

### B.2. DiffCSP++

For DiffCSP++, we used the hyperparameters specified by the original authors for the MP20 dataset.

### B.3. SymmCD

For SymmCD, we used the hyperparameters specified by the original authors for the MP20 dataset, except for the number of training epochs and batch size, which we reduced to 1 000 and 256, respectively, to ensure fair comparisons.

When generating materials using SymmCD, we encountered an issue where the length and angle matrices contained NaN, Inf, or extremely small values. To facilitate subsequent evaluation, we filtered out those invalid materials.

## C. Wren Energy Histograms

We show the similarities of generated material distrubution across all model versions WYCKOFFDIFF-marginal, WYCK-OFFDIFF-uniform, and WYCKOFFDIFF-zeros, in Figure 4.

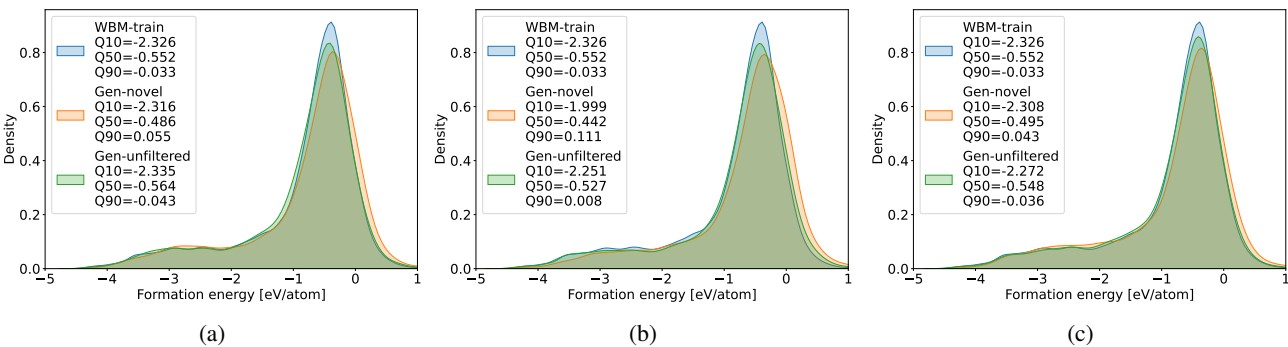

*Figure 4.* Distribution of formation energies predicted by Wren for, (unfiltered) generated protostructures, novel generated protostructures, relative to the training set for the model. Protostructures are generated by (a) WYCKOFFDIFF-marginal (b) WYCKOFFDIFF-uniform (c) WYCKOFFDIFF-zeros. Q10, Q50,and Q90 are the 10th, 50th, and 90th percentiles respectively.

---

[9]https://github.com/txie-93/cdvae
[10]https://github.com/jiaor17/DiffCSP-PP/
[11]https://github.com/sibasmarak/SymmCD

*Table 4.* Results on the Carbon24 dataset. Due to overall very low novelty, we settled for only 1 000 protostructures, and 1 000 novel protostructures, for computing statistics.

| | Model | FWD $\downarrow$ | Nov. $\uparrow$ (%) | Uniq. $\uparrow$ (%) | Novel FWD $\downarrow$ | Novel Uniq. $\uparrow$ (%) | Nov./min. $\uparrow$ |
|---|---|---|---|---|---|---|---|
| | CDVAE | $110 \pm 5.62$ | $5.30 \pm 1.45$ | $8.4 \pm 0.8$ | $91.6 \pm 7.5$ | $16.7 \pm 2.10$ | 3 |
| | DiffCSP++ | $4.12 \pm 1.53$ | $1.40 \pm 0.46$ | $16.6 \pm 0.60$ | $38.6 \pm 4.93$ | $22.3 \pm 1.47$ | 2 |
| | SymmCD** | $11.4 \pm 1.85$ | $6.53 \pm 1.72$ | $16.4 \pm 0.46$ | $94.8 \pm 33.1$ | $21.7 \pm 3.84$ | 6 |
| OURS | WyckoffDiff-uniform | $0.78 \pm 0.14$ | $1.6 \pm 0.87$ | $19.0 \pm 5.69$ | $52.9 \pm 8.34$ | $23.8 \pm 2.93$ | 14 |
| | WyckoffDiff-marginal | $0.78 \pm 028$ | $1.4 \pm 0.47$ | $16.4 \pm 0.89$ | $53.0 \pm 2.76$ | $29.0 \pm 3.42$ | 14 |
| | WyckoffDiff-zeros | $0.89 \pm 0.21$ | $1.6 \pm 0.40$ | $16.2 \pm 0.55$ | $49.0 \pm 4.41$ | $27.8 \pm 1.95$ | 12 |

*Table 5.* The number of unique and novel prototypes among 1 000 novel protostructures from models trained on the Carbon24 dataset. Due to overall very low novelty, we settled for only 1 000 novel protostructures for computing statistics.

| Model | # Unique & Novel Prototypes |
|---|---|
| CDVAE | $167 \pm 21$ |
| DiffCSP++ | $223 \pm 15$ |
| SymmCD | $217 \pm 38$ |
| WyckoffDiff-unif. | $237 \pm 25$ |
| WyckoffDiff-marg. | $290 \pm 48$ |
| WyckoffDiff-zeros | $278 \pm 15$ |

## D. Validity of Materials

Other related works (e.g., CDVAE (Xie et al., 2022) and subsequent works) present two metrics on "validity" of materials.

**Structural validity** A material is determined to be *structurally* valid if the distance between two atoms is less than 0.5 Å. As we are only concerned with protostructures (and thus do not consider the exact geometries), this metric is not applicable in our study.

**Compositional validity** If a materials has an overall neutral charge according to SMACT (Davies et al., 2019), it is determined to be *compositionally* valid, which is something that can be computed for protostructures. When computing this on the novel protostructures, this number is $81.8 \pm 0.3\%$ for CDVAE, $87.1 \pm 0.51\%$ for DiffCSP++, $86.3 \pm 0.28\%$ for SymmCD and $85.9 \pm 1\%$, $87.7 \pm 0.4$, and $86.1 \pm 0.3\%$ for WyckoffDiff-uniform, WyckoffDiff-marginal, and WyckoffDiff-zeros, respectively. However, the term "validity" in this case should not be taken as a prerequisite for a real material, as some systems do not fulfill this (e.g., metals with diffuse non-local bonds). Indeed, the validity of the materials in WBM is 87%, and it is hence not expected (nor desirable) to have this number any higher.

## E. Results on Carbon24

As an additional experiment, we used Carbon24 (Pickard, 2020). We used the same training set as in the DiffCSP++ repository, and all baselines used hyperparameter configurations from their corresponding repositories (see above). We used the same hyperparameters for WyckoffDiff as for WBM, apart from training for 4 000 epochs to match the baselines, and setting $P = 24$.

However, all models strggled in general to generate novel protostructures, probably due the dataset containing only a single element, and a novel protostructure hence needed to be a novel prototype. due to this low novelty, we limited the study to compute statistics on 1 000 protostructures, and 1 000 novel protostructures. We present the numbers in Tables 4 and 5.

## F. Novel Stable Materials

Low formation energies are only indirectly related to stability; the thermodynamically stable material at a composition is the one with the lowest formation energy compared to all alternative competing phases and linear combinations of phases, which spans the so called convex hull of thermodynamical stability (see, e.g., (Bartel et al., 2020)). I.e., in order do determine if a novel material is stable, the formation energy needs to be compared with the convex hull. Deriving the formation energy of a material and computing the convex hull is described below.

### F.1. Formation Energy

*Formation energy* is calculated by taking the total energy of a material and subtracting the sum of elemental solid energy for each element present in the material. A negative formation energy therefore implies a lower energy state of the material relative to its elemental components. In turn, the formation energy proves that the material will not decompose into its elemental components.

### F.2. Convex Hull

Plotting the formation energies of the materials and its corresponding elemental solid energies in a diagram constructs a *phase diagram*. Materials that holds the lowest formation energy in the phase diagram forms a *convex hull*. The convex hull constructs serves as the line of stable materials, meaning: if a new crystal structure is discovered but has higher formation energy in comparison to the convex hull, the new crystal structure will decompose into its closest stable neighbors on the convex hull; whereas if the new crystal structure has a lower formation energy in comparison to the convex hull, the new material is novel and stable. The novel stable material is then part of a new convex hull, redefining the line of stable materials.

## G. Supplementary Details on Materials Discovery Demonstration

### G.1. Additional Protostructures

As described in Section 5 we performed a selection of three chemically interesting materials, whereas it was noted that there where a total of eight fluorides with distinctly below the convex hull. In Table 6 we list the materials sorted on energy distance from the convex hull of WBM and Materials Project (MP), up to the final selected structure.

*Table 6.* Listed structures up to the final included selection of interesting chemistry. The top section is the eight fluorides with formation energy distinctly below the convex hull. † Selection of a three examples with interesting chemistry out of WYCKOFFDIFF generated crystal structures close to or below the convex hull of WBM and Materials Project (MP).

| Protostrucuture | E form. $[eV/atom]$ | E above hull $[eV]$ |
|---|---|---|
| AB6C_hR24_166_a_h_b:Cs-F-Sn † | −2.6103 | −0.0322 |
| A2B6CD_cF40_225_c_e_a_b:Cs-F-Ni-Rb | −2.6043 | −0.0194 |
| AB6C_hR24_148_a_f_b:Ba-F-W | −3.2550 | −0.0097 |
| A6BC_cF32_225_e_a_b:F-Li-Ru | −2.4313 | −0.0076 |
| A6B3C_mC20_12_ij_ai_d:F-Rb-V | −3.1621 | −0.0068 |
| A5B2C_tP8_123_bj_e_a:F-K-Zn | −2.6064 | −0.0038 |
| A6BC_cF32_225_e_a_b:F-K-Ta | −3.6294 | −0.0027 |
| A6BC_mC16_12_ij_a_d:F-Ti-Zn | −3.3484 | −0.0019 |
| ABC2_hP8_194_a_c_f:Na-Nb-O † | −2.5369 | −0.0009 |
| A6BCD2_cF40_225_e_a_b_c:F-Ga-Na-Rb | −3.0972 | −0.0003 |
| ABC4_tI24_141_a_b_h:As-Nd-O | −2.8137 | −0.0003 |
| A3B_hR24_167_e_b:F-Ga | −2.9513 | −0.0002 |
| A2BC7D2_hR36_155_c_a_bf_c:Al-Ba-O-Sb | −2.7786 | $-3.07 \times 10^{-05}$ |
| AB_cF8_225_a_b:Ca-O | −3.3142 | $6.16 \times 10^{-05}$ |
| ABC_hP6_194_c_d_a:F-La-Se | −3.1550 | $6.20 \times 10^{-05}$ |
| AB4C_oC24_63_c_fg_c:Ca-O-S | −2.6801 | $8.25 \times 10^{-05}$ |
| A2BC_hR12_166_c_a_b:Ca-I-P † | −1.4222 | 0.0001 |

### G.2. Density Functional Theory Supplementary Details

In order to maintain compatibility with MP and WBM dataset, all DFT calculations and post-corrections (`MaterialsProjectCompatibility`) (Jain et al., 2011) were performed using INCAR settings, KPOINTS and pseudo-potentials defined by Pymatgen (Ong et al., 2013). Calculations where converged to atleast 1e-4 eV in total energy in electronic steps.

### G.3. Previous WYCKOFFDIFF-marginal

The structure `A2BC_hR12_166_c_a_b:Ca-I-P` ($Ca_2PI$) was found using a previous iteration of the WyckoffGNN architechure where we did not use softmax in the message-function $M_l$, but instead used the raw outputs of the neural network (see Algorithm 4), and encoded the degrees of freedom of the position using a binary representation, i.e., constrained or unconstrained position instead of 0, 1, 2, or 3 degrees of freedom.

