# OpenReview forum: "WyckoffDiff -- A Generative Diffusion Model for Crystal Symmetry"
_ICML.cc/2025/Conference — ICML 2025 poster_

### Official Review · Reviewer_Co5n · 2025-03-13

**Overall Recommendation:** 2

**Summary:**

The authors introduced WyckoffDiff, a discrete diffusion model for generating crystalline materials with explicit symmetry constraints. WyckoffDiff encodes crystals via protostructure, which includes a space group and Wyckoff positions. The authors evaluated their approach on materials benchmark (WBM dataset). They also proposed a new evaluation metric, the Fréchet Wrenformer Distance (FWD), which quantifies how closely the generated protostructures match the symmetry characteristics of the training data (analogous to FID).

**Claims And Evidence:**

Yes.

**Essential References Not Discussed:**

No.

**Experimental Designs Or Analyses:**

Yes.

**Methods And Evaluation Criteria:**

Yes.

**Other Comments Or Suggestions:**

No.

**Other Strengths And Weaknesses:**

**Strengths**:

- Integration of crystal symmetry:
The idea of explicitly incorporating crystallographic symmetry into the generative modeling process via discrete diffusion models is novel. The authors introduce a "protostructure" representation that simplifies the crystal generation problem by explicitly encoding symmetry constraints, potentially enabling efficient material generation.
- Evaluation and metrics:
The authors introduced a new evaluation metric, Fréchet Wrenformer Distance (FWD), that is designed to evaluate the quality of the generated materials by measring the similarity with the training set.

**Weaknesses**:

The paper still has several weaknesses.

- Unclear differences and advantages from existing methods: While the paper integrates crystal symmetry, the differences from existing symmetry-aware methods, such as SymmCD and WyCryst is not clear. For instance, Wyckoff also incorporates crystal symmetry through Wyckoff positions. The paper did not provide a sufficiently detailed comparison of the differences and advantages of WyckoffDiff over these methods.

- Converting protostructures to atomic structures:
While generating protostructures simplifies the generation step, the practical utility depends heavily on the subsequent realization into full atomic structures. The realization step might involve additional complexity (e.g., choosing internal degrees of freedom) that can affect the final material properties. The paper did not address or quantify how protostructures can be converted into a full crystal structure.

**Questions For Authors:**

Can you explain if is possible to convert the generated protostructures into complete atomic geometries?

**Relation To Broader Scientific Literature:**

The idea is built on discrete diffusion models (D3PM) to generate the protostructures.

**Theoretical Claims:**

No.

---

> ### Author Rebuttal · Authors · 2025-04-01
>
> We are happy to see that the reviewer thinks our approach is novel and that generating protostructures is an approach that can enable efficient material generation. We address the concern that we do not quantify how structures can be realized from protostructures, and elaborate on differences between related work, below.
>
> ## We do address how protostructrues can be realized into full atomic structures, run a proof-of-concept study, and discover new materials
> In section 5 (with further details in the appendix), we presented a study on how the generated protostructures can be realized into full structures, and following this simple approach, we also find materials outside of the WBM dataset that are below the convex hull (see fig. 3 for illustrations of the crystal structures and the corresponding phase diagrams), showcasing the effectiveness of WyckoffDiff in generating protostructures which subsequently can also be realized into stable materials. The aim of this study was to highlight that while protostructures alone are not enough, it is relatively straightforward to obtain full structures and doing this, we do actually find new materials, showcasing the usefulness of WyckoffDiff for materials discovery. We will add an additional mention of this to the abstract.
>
> ## We will elaborate on differences to WyCryst and SymmCD
> We agree with the reviewer that a further elaboration on the differences between our method can make the paper clearer. We will add this to the related work section, but to summarize:
>
> * **WyCryst:** This work is similar in the sense that it generates a Wyckoff-based description of the material (and then assign exact coordinates in a later step). However, it is based on a different Wyckoff-representation, and they do not discuss how the problem of positions with 0 degrees of freedom (in our work called constrained positions) are handled. This last point is something that we are very careful to respect, and the reason for why we separate the positions into "constrained" and "unconstrained". We can also not find that the describe how they handle the varying number of Wyckoff positions for different space groups (which we do by using our graph representation). Their work also focus on generating strictly ternary materials while we put no such restrictions on the materials.
>
> * **SymmCD:** Apart from also generating positions, SymmCD takes on a different approach when assigning Wyckoff positions: essentially it starts by sampling a number M of "representative" atoms which is kept fixed, and then the element, Wyckoff position, and multiplicity of these respective atoms are generated (these atoms are representatives of the corresponding orbit). We, on the other hand "start" from all Wyckoff positions, and then generate which element(s) (if any) occupy each position, and their respective multiplicity, which is conceptually different. Given the results in sec 4, it also seems very effective.

---

### Official Review · Reviewer_j5np · 2025-03-16

**Overall Recommendation:** 3

**Summary:**

In this paper, the authors propose a symmetry-aware generative model for crystal generation. The proposed Wyckoff diffusion model generates a protype based on elements taking Wychoff positions instead of 3D positions defined in a unit cell like a few methods in literature. They show that using this with a discrete diffusion model and Frechet Wrenformer distance (FWD) for evaluation leads to a more expressive generative model generating diverse samples (wrt to symmetries) for crystal generation. Material Project dataset is used for benchmarking with other models on generation task.

## post rebuttal
I have read the rebuttal. The authors provide a satisfactory rebuttal and thus I keep my score.

**Claims And Evidence:**

Yes, the claims in the paper are clear.

**Essential References Not Discussed:**

Yes, most of the related works are covered in the paper.
Missed reference in related works.

1. WyCryst: Wyckoff inorganic crystal generator framework, Zhu et al. [Matter 24]
2. FlowMM: Generating Materials with Riemannian Flow Matching, Miller et al. [ICML 24]

A few sentences in the paper seem to require citations that are missing. Listing them below:

ln 31 [column 2]: 'Additionally, the infinite space of continuous coordinates also
opens the risk of generating degenerate materials or structures outside of the symmetry proximity. '

This does not seem to be the case for FlowMM or DiffCSP, which uses E(3) symmetries using continuous coordinates on a torus or Euclidiean plane.

ln 34 [column 2]: 'Since materials of
high symmetry are generally the interesting materials to explore, generation of large sets of low symmetry materials is
inefficient.'

ln 85 [column 2] typo in AFLOW prototype label

**Experimental Designs Or Analyses:**

Yes.

The proposed model in the paper generates protostructures and evaluates the novelty of the protostructures generated. Although given that other methods do have a protostructure-based model, it would be useful to have the usual evaluation metric for a few generation samples of each protostructure to have a fair comparison.

**Methods And Evaluation Criteria:**

The proposed methods and evaluation criteria are correctly done. Missing a few commonly used evaluation metrics for crystal generation:
 like validity, coverage, stability, cost, SUN rate, and SUN cost.  See references [2,5].

2. SymmCD: Symmetry-Preserving Crystal Generation with Diffusion Models, Levy et al. [ICLR 25]

5. FlowMM: Generating Materials with Riemannian Flow Matching, Miller et al. [ICML 24]

**Other Comments Or Suggestions:**

It would be interesting to see additional material datasets in the experiment section, like Carbon24 or Perov5.

**Other Strengths And Weaknesses:**

## Strengths
- The related work section is well written, and relatively new ideas like Wyckoff positions and Wren energies are explained well.
- The proposed method of combining wchkoff positions modeling with discrete diffusion seems to be a reasonable choice for generating crystals.

## Weaknesses

- Missing a few commonly used evaluation metrics for crystal generation: like validity, coverage, stability, cost, SUN rate, SUN cost, see references [2,5].
- Additional dataset for experiments would have been useful.

**Questions For Authors:**

1. ln 34 [column 2]: 'Since materials of
high symmetry are generally the interesting materials to explore, generation of large sets of low symmetry materials is
inefficient.' What is low symmetry?

2. What is the scalability of the proposed method?

3. ln 322[ column 2] 'compared to baselines..' The explanation provided for lower novelty for Wyckoffdiff is unclear.

4. Do the numbers reported in Table 1 exclude the void materials with 0 atoms?, Is it the same for SymmCD, where the NaN values are removed?


and see Weaknesses.

**Relation To Broader Scientific Literature:**

The key contributions of the paper on using Wyckoff positions are not entirely novel (see reference [1]). Previously, space groups of crystals have been used [2] in addition to modeling atom positions, while [3,4,5] use E(3) symmetries for modeling crystals. Using the symmetries commonly found in crystals through Wyckoff position seems to be a reasonable choice and combining it with discrete diffusion is also useful.

1.  WyCryst: Wyckoff inorganic crystal generator framework, Zhu et al. [Matter 24]

2. SymmCD, Levy et al. [ICLR 25]

3., 4.  DiffCSP and DiffCSP++, Jiao et al.[NeurIPS 23]

5. FlowMM: Generating Materials with Riemannian Flow Matching, Miller et al. [ICML 24]

**Theoretical Claims:**

There are no theoretical claims made in the paper.

---

> ### Author Rebuttal · Authors · 2025-04-01
>
> We greatly appreciate the comments from the reviewer that have helped improving the clarity of the paper and improved the numerical evaluation. We are happy that the reviewer agrees with us that using discrete diffusion for generation of materials based on Wyckoff positions is a reasonable and useful approach. We address the comments related to evaluation and make some clarifications below.
>
> ## We have now computed more metrics compatible with protostructures
> It should first be noted that since we are not generating full geometrical information in our method, metrics that require this are generally not accessible to us which applies to several metrics used in previous works. Hence, we have relied on FWD and others we can compute (Table 1), and statistics on prototypes (Table 2).
>
> This also applies to the SUN metric proposed by the reviewer. We therefore adapted a protostructure version of SUN, SUN-Wren, where stability is computed by using the Wren model to estimate formation energies directly from the _protostructures_, which is then compared to the Materials Project (MP) 2023 convex hull, as for stability in FlowMM. For novelty, we compared relative to the _protostructures_ present in the training set WBM. Uniqueness can similarly be compared to other _protostructures_ in the generated set. We can then also calculate the stability ratio of our training set WBM relative the MP 2023 hull to compute a reference stability score that arguably is what all generative models trained on this data set should strive to match. We only apply this metric on structures containing elements with atomic number 1 to 83 as this is where Wren has been trained for. We find that the symmetry-based models obtain SUN-Wren ratio close to WBM while CDVAE produces protostructures that are to a much larger degree unstable. We think this emphasizes the importance of incorporating symmetry in the models, as this helps producing useful materials (see next section). Due to the short time frame in which we create these, they should be regarded as preliminary and we may need to update them in further responses (if the referees agree it is a relevant metric).
>
> ## Low symmetry materials are not the interesting materials found in databases (related to line 31 and 34, col 2)
> Our notion of symmetry starts from classifying crystal structures into space groups, which are defined by a set of symmetry operators that map the crystal structure on itself. Space group 1 has the lowest symmetry, as there is no symmetry at all. See comment to pC7E for more on why higher symmetry is the interesting materials. In addition to that answer, FlowMM learns distributions that are invariant to SE(3) transformations, which DiffCSP++ (which is extension of DiffCSP) and SymmCD also do, but they additionally incorporate knowledge about space group symmetries, and that is why we benchmark against these.
>
> ## We investigate additional dataset
> As suggested, we ran experiments also on Carbon24, as we are mostly interested in the structural properties of materials and Perov5 only contains Perovskites which is a certain fixed structure. We find that all models in general struggle to generate novel protostructures: the novelty for DiffCSP++ and WyckoffDiff is a mere 1-2 \%, while for SymmCD the number is 6.5 \%. However, WyckoffDiff is again much faster, with 12-14 nov/min compared to second best SymmCD with 6 nov/min, and places in between DiffCSP++ and SymmCD in FWD(novel). CDVAE has novelty of $\sim$ 7 \%, but again has a much higher FWD than the other models.
>
> ## We will clarify regarding low novelty
> We think the low novelty arises due to "memorization" from the model, which is why we present the results of training for shorter times, showing that this increases the novelty. We will add a clarification about that in line 315 col 2. See also response to pC7E related to novelty.
>
> ## Regarding scalability
> A bottleneck in our method is that we are operating on complete graphs, meaning that for space groups with many positions, the number of edges in the graph increases quickly. On the other hand the data dimensionality is fixed for a certain space group, and more atoms in the unit cell does not change that. E.g., in fig 1, the number of Cs atoms occupying the "c" position is represented by an integer, so increasing this from 0 to, say, 4, doesn't affect the dimensionality of the data. Increasing the size of the set of elements in the materials will make the unconstrained positions grow in size, but for WBM we without problem used 100 as the maximum atom number which is high.
>
> ## Numbers in Table 1 exclude void and NaN materials
> Materials are filtered before computing metrics, but we still make sure to generate enough materials to have 10k materials after filtering. We will change the text for the two stars ** in the caption of Table 1 to clarify this.
>
> ## We will add references to WyCryst and FlowMM
> See also discussion about WyCryst in response to Co5n.

---

### Official Review · Reviewer_pC7E · 2025-03-17

**Overall Recommendation:** 2

**Summary:**

The paper introduces WYCKOFFDIFF, a novel framework for generating crystal structures using a discrete diffusion process that inherently preserves symmetry. By representing crystal protostructures based on Wyckoff positions, the method partitions these positions into constrained (fixed) and unconstrained (flexible) categories, effectively encoding the intrinsic symmetry of crystals. The framework adopts the D3PM generative process alongside a novel WyckoffGNN backbone to model the occupancy of each Wyckoff position, enabling both symmetry-aware generation and rapid sampling. Additionally, the paper proposes the Fréchet Wrenformer Distance, a symmetry-sensitive metric for evaluating the quality of the generated structures. The authors finally validate model generation results by realizing the crystal structure to actual materials in the lab.

**Claims And Evidence:**

The authors claim their method enjoys symmetry-aware generation and fast sampling. The claims are supported by the FWD score and Nov./Min respectively in Table 1.

**Essential References Not Discussed:**

In the reviewer’s knowledge, no.

**Experimental Designs Or Analyses:**

The experimental design, as presented in Tables 1 and 2, is generally well-structured. The authors evaluate the crystals generated by their model and compare them with existing methods using metrics such as FWD, novelty, uniqueness, and sampling speed. These evaluations provide insight into the quality of the generated structures and support the authors’ claims regarding symmetry-aware generation.
However, some experimental results seem to challenge rather than reinforce the proposed framework. For instance, in Table 1, all variants of WyckoffDiff perform significantly worse than CDVAE in terms of novelty. The authors attempt to justify this by attributing it to training time and arguing that faster sampling compensates for the lower novelty, but this explanation lacks a rigorous analysis. Additionally, the authors define a high FWD score as an indication that the generated structures deviate significantly from the training distribution. While CDVAE has a high FWD, it also achieves a notably high novelty score, suggesting that it may be generating novel and valid crystals.
To strengthen their claims, the authors could incorporate a validity metric. If they can demonstrate that CDVAE produces novel but largely invalid structures, it would provide stronger support for WyckoffDiff’s advantage in generating physically meaningful crystal structures.
Also, the paper lacks an ablation experiment on Wyckoff position encoding. To showcase the benefits of encoding symmetry with Wyckoff positions, an experiment on generating crystals with D3PM alone without Wyckoff positions should be needed.

**Methods And Evaluation Criteria:**

The proposed method, which encodes crystal structures using Wyckoff positions and employs the WyckoffGNN, is specifically designed to capture the intrinsic symmetry of crystals, making it well-suited for the problem at hand. The authors evaluate their model on the WBM dataset, a well-established crystal structure dataset.

**Other Comments Or Suggestions:**

No.

**Other Strengths And Weaknesses:**

Strengths: Encoding symmetry with Wyckoff positions is both original and interesting.

Weakness: The benchmarking is not rigorous enough to convincingly demonstrate the proposed framework’s performance.

**Questions For Authors:**

I summarize my questions as below (See Experimental Designs or Analyses for more details):
1. Do the authors have a more convincing explanation on the poor performance on novelty as in Table 1 and 2?
2. Have the authors conducted an ablation experiment on encoding Wyckoff positions?

If both concerns are fully addressed, that is the authors show WyckoffDiff generates both meaningful and novel samples and the proposed Wyckoff method is indeed beneficial , I will change my evaluation.

**Relation To Broader Scientific Literature:**

Prior methods lack consideration of symmetry. This paper explicitly models symmetry with Wyckoff positions.

**Theoretical Claims:**

There is no proof.

---

> ### Author Rebuttal · Authors · 2025-04-01
>
> We are very happy to see that the reviewer thinks our method is both original and interesting. We address the concerns related to the comparison to CDVAE and the proposed ablation study below.
>
> ## While generating many novel structures, CDVAE does not generate symmetrical materials, which are the most interesting
> First, as the reviewer rightly points out, attaining a high novelty in itself is not enough (nor difficult), unless the materials are also at least "valid" and have interesting properties. We discuss validity in a later section. Regarding what materials are interesting: a significant portion of known crystalline materials, such as metals, salts, intermetallics, and covalent solids like diamond, silicon, etc., are built from small unit cells with ordered arrangements of atoms in a high degree of symmetry. As discussed in the introduction of our paper, this is also usually reflected in databases of materials, where one often sees a high representation of materials with space groups of higher symmetry. Hence, it is ultimately a (not uncommon) choice in this work to focus on materials in this category for materials discovery (which is not saying that this is the _only_ interesting category). If one do not enforce high symmetry in a generative model, it has to learn this property from data, which makes it more difficult to appropriately reproduce this aspect.
>
> In the initial submission, we evaluated the quality of the generated materials using FWD, where CDVAE has a much higher FWD than the alternative methods, also when **computed only on the novel materials**. This suggests that CDVAE does not generate materials that respect the symmetries of the training set. This can further be assessed by considering the space group distributions: 36 \% of the materials generated by CDVAE fall into space group 1, i.e., no symmetry, and $>90$ \% of the materials belong to spacegroups 1-15 (the lowest symmetry groups). The corresponding numbers for the WBM dataset (the training split used in our paper) are 0.3 \% (SG 1) and 13 \% (SG 1-15). As we sample from the empirical SG distribution, we will follow this distribution by construction. We appreciate that the reviewer brings this up, and we will add this discussion to line 315 col 2 to further elaborate on the numbers for CDVAE (apart from the FWD), and add the numbers related to space groups.
>
> Additionally, in our opinion, it is a less important feature for these methods to have a very high novelty, as long as it is reasonably large. It is trivial to extend any generative method with a final “novelty filter” step (essentially taking a rejection sampling approach, only keeping novel materials). This is arguably the most reasonable approach in a materials discovery setting to not spend resources on duplicates of training data, and thus ensures a novelty of 100\% by construction. Hence, the “raw novelty” is less relevant, and the more important metric is the novel materials/minute after a rejection sampling step. Using this criterion, we see that WyckoffDiff performs much better than CDVAE (119-159 vs 71 Nov/Min; see rightmost column in Table 1), in addition to having a much better FWD on the filtered set of (only novel) generated materials. See response to j5np regarding the novelty of WyckoffDiff.
>
> ## Removing encoding Wyckoff positions would not generate protostructures
> The task we are tackling is the generation of protostructures, which essentially contains information of space group, elements, and which Wyckoff position each element occupy. Performing an ablation where we use **D3PM without encoding the Wyckoff positions would hence result in only generating a set of atoms, from which it is not possible to construct a protostructure**. In other words, removing the Wyckoff positions from the representation changes the problem. We are therefore unsure what the reviewer means when suggesting this ablation, but are happy to answer any follow up comment.
>
> ## We have now looked at validity
> We took the advice from the reviewer about validity metrics, and computed the compositional validity metric based on SMACT used in DiffCSP++ on the novel materials. This number is 84-85 \% for WyckoffDiff, while it is 81-82 \% for CDVAE. However, this metric can be misleading if interpreted as a measure of the number of "valid generations" since it is based on a typical matching of the charge states of ions in a material which often, but not always, is fulfilled by actual systems (e.g., metals with diffuse non-local bonds). Indeed, the training set we use for our models, WBM, only has roughly 87\% of this "validity". In other words, it is not expected (nor desirable) to have this number any higher. We are open to further suggestions of useful validity metrics to explore.

---

### Official Review · Reviewer_btpV · 2025-03-18

**Overall Recommendation:** 3

**Summary:**

This paper proposes a novel generative model, WyckoffDiff, for generating Wyckoff representations of crystal materials. By applying discrete diffusion models directly to Wyckoff representations, WyckoffDiff can generate diverse protostructures for crystal materials. Additionally, WyckoffDiff leverages GNN models to derive Wyckoff representations through message passing over a complete graph.

## update after rebuttal
After reviewing the rebuttal and considering the comments from other reviewers, this reviewer remains concerned about the limitations of the proposed approach. While generating protostructures is certainly a viable strategy, it necessitates additional postprocessing to obtain full-atom structures. Moreover, the paper would benefit from clearer articulation of its motivation, more thorough comparisons with existing methods, and additional details regarding certain implementation steps. As such, the reviewer currently leans toward maintaining the original score.

**Claims And Evidence:**

Yes, the claims are well-supported by the empirical evidence.

**Essential References Not Discussed:**

There are no critically essential related works missing from this paper. While some may exist, they can be incorporated later.

**Experimental Designs Or Analyses:**

From the perspective of protostructure generation, the empirical design choices appear reasonable. However, the experimental results primarily compare previous baselines in the context of protostructure generation rather than full geometry generation, even though the baseline methods were originally designed for the latter. This raises concerns about how these baselines were adapted for protostructure generation. Without proper adaptation, the comparison may be misleading or unfair.

**Methods And Evaluation Criteria:**

The proposed method is well-suited for the crystal material generation problem, and the chosen evaluation metrics are appropriate.

**Other Comments Or Suggestions:**

(1) The reviewer suggests incorporating more appropriate baselines. While it is understandable that existing works rarely follow the same problem formulation, the current evaluation of the proposed method primarily relies on showcasing newly generated samples that fall below the convex hull (as measured by oracle models). However, this approach is not entirely reliable.

(2) The current empirical results are not well-structured. In Table 1, WyckoffDiff does not demonstrate clear superiority in terms of novelty or uniqueness compared to the selected baselines. The reviewer suggests that the authors should at least highlight the best-performing model for each metric and consider whether all reported variations are necessary.

**Other Strengths And Weaknesses:**

Strengths:

(1) The paper is well-written, and the motivation behind the idea is clearly articulated.

(2) Incorporating symmetries is crucial for efficiently generating highly symmetric crystal materials.

(3) Applying discrete diffusion for generating Wyckoff representations is a reasonable approach.

Weaknesses:

(1) The empirical comparisons are currently insufficient, as the selected baseline methods are limited. Additionally, the proposed method does not demonstrate clear superiority over the baselines. The reviewer suggests that the comparison should focus on generating complete crystal symmetries, as this is the ultimate goal. Evaluating only Wyckoff generation is not entirely convincing.

(2) A major limitation of WyckoffDiff is that it cannot directly generate the final crystal geometry. Instead, it only produces intermediate Wyckoff representations, requiring additional steps to obtain the complete crystal structure.

**Questions For Authors:**

The primary limitation of the proposed approach appears to be its inability to generate the full geometry of crystal materials.

(1) How are the baseline methods adapted for protostructure generation?

(2) Is protostructure generation sufficient for material design? Given that the ultimate goal is to generate complete material geometries, it seems that the proposed WyckoffDiff model falls short in this aspect.

**Relation To Broader Scientific Literature:**

The key contributions of the paper are mainly related to discrete diffusion models and crystal material generation.

**Theoretical Claims:**

This paper does not contain theoretical claims.

---

> ### Author Rebuttal · Authors · 2025-04-01
>
> We thank the reviewer for their comments, and we are happy to see that the reviewer agrees that incorporating symmetries is crucial for efficient material generation, and that our approach of applying discrete diffusion for working with the Wyckoff representation is reasonable. Below, we address and try to clear out the questions and concerns raised.
>
> ## We have compared with very competitive baseline methods for material generation, without modifications
> The motivation for the paper is that materials typically exhibit high symmetry, and we develop a method which incorporates that explicitly. We have compared with very competitive material generation baselines which span not incorporating symmetry (CDVAE), respecting symmetry but relying on templates (DiffCSP++), and respecting symmetry without relying on any templates (SymmCD). While these do not tackle the generation of protostructures explicitly, any generated material will of course _have a protostructure_, and a good method for material generation should also be able to generate materials with proper protostructures. Note that we did not modify the baseline methods, but simply computed the protostructures from the generated crystal structures (line 296 col 2). The reviewer do not specify which other baselines they think should be compared with (they also say that _"There are no critically essential related works missing"_), but the two other baselines mentioned in other reviews are FlowMM and WyCryst. The former does not explicitly enforce symmetry and would thus fall in the same category as CDVAE, and while the latter is conceptually similar to our, it faces some issues related to handling positions with 0 degrees of freedom (which we discuss in the response to Co5n). Also, the public code does not include code for unconditional generation which is the task we are facing, and the paper does not describe the method well enough for us to implement it ourselves (for example, the underlying neural network is not described, and generation is not described how it is done in practice, only training).
>
> ## We use various metrics in our evaluation, not only stable materials
> We respectfully disagree with the reviewer that we primarily rely on showcasing newly generated samples that fall below the convex hull. The comparison of realized structures against the convex hull is a proof-of-concept study which highlights that out method can indeed generate practically useful structures. This is not part of the comparison against other methods. In sec 4, however, we have tried to showcase various metrics which quantitatively highlight different aspects of the generated protostructures, and we have now added additional (see response to j5npy) such metrics. It is difficult to evaluate discrete generative models, as for example a metric such as uniqueness will not be $100 \%$ even for a "perfect" model. From a materials discovery perspective we would like to have high novelty, but as we also write in the response to pC7E, we can obtain 100\% novelty for all models by applying a filter that removes all non-novel materials, and then it is the time per generation that becomes the important metric, which is why we include the nov./min. metric. We will extend the discussion related to Table 1, including these aspects, in a revised version of the manuscript.
>
> We think that reporting variations in metrics is very important to show the sensitivity of the methods.
>
> ## Generating protostructures is an intermediate step, and we demonstrate how full geometries can be obtained
> It is true that generating protostructures is an intermediate step, but as we show in section 5, full geometries can be obtained from them. As we discuss in section 6, separating these processes opens up the possibility to both do additional filtering of the protostructures to focus on the most promising ones and hence use resources where they are most needed, but also use specialized methods for the different steps. Hence, this approach introduces more flexibility for using methods tailored for respective task (e.g., obtaining the positions by using a large pretrained ML potential for DFT relaxation (as we do in sec 5) instead of training a diffusion model. While full structures is the end goal, dividing the process into multiple steps is a novel design and a strength in our model that offers unique advantages compared to the other methods. By the successful demonstration of the method as useful for materials design in the proof-of-concept study in sec 5, we think this can inspire continued development and improvements also by others.

---

### Decision · Program_Chairs · 2025-05-01

**Decision:**

Accept (poster)

**Comment:**

This papers presents a discrete diffusion model to generate crystal structures based on the symmetry properties of the crystal. The model employs a representation that includes the elements, the space group and the Wyckoff positions occupied by the atoms. The generated samples do not fully describe a crystal but the authors offer a post-processing procedure to obtain atomic positions, by obtaining initial positions with PyXtal and then relaxing the structure with MACE. The main criticism by reviewers is that the model generates proto-structures instead of fully described crystals. However, in my opinion, the authors provide convincing arguments both in the rebuttal and in the paper about why this is not necessarily a limitation. While the results are not spectacular and the reviewers are not overly enthusiastic, the reviews do not reflect any fundamental flaws and I also did not find any in my reading of the paper. Furthermore, all reviewers agree that the paper is well written. In this regard, I think the paper has sufficient quality and would be appreciated by at least some members of the community and serve as inspiration for future work. For these reasons, I recommend the acceptance of the paper.